# Amyloid-Beta Peptides and Activated Astroglia Impairs Proliferation of Nerve Growth Factor Releasing Cells In Vitro: Implication for Encapsulated Cell Biodelivery-Mediated AD Therapy

**DOI:** 10.3390/cells10112834

**Published:** 2021-10-21

**Authors:** Sumonto Mitra, Silvia Turchetto, Winant Van Os, Lars U. Wahlberg, Bengt Linderoth, Homira Behbahani, Maria Eriksdotter

**Affiliations:** 1Division of Clinical Geriatrics, Center for Alzheimer Research, Department of Neurobiology, Care Sciences and Society (NVS), Karolinska Institutet, 141 52 Stockholm, Sweden; Maria.eriksdotter@ki.se; 2Division of Neurogeriatrics, Center for Alzheimer Research, Department of Neurobiology, Care Sciences and Society (NVS), Karolinska Institutet, 171 64 Stockholm, Sweden; silvia.turchetto@uliege.be (S.T.); w.l.van.os@lic.leidenuniv.nl (W.V.O.); 3GIGA-Stem Cells and GIGA-Neurosciences, Interdisciplinary Cluster for Applied Geno-Proteomics (GIGA-R), University of Liège, 4000 Liège, Belgium; 4Wiskunde en Natuurwetenschappen, LIC/Chemical Biology, Leiden Institute of Chemistry, 2300 RA Leiden, The Netherlands; 5Gloriana Therapeutics, Inc., Warren, Rhode Island, RI 02885, USA; LUW@GlorianaTx.com; 6Department of Clinical Neuroscience, Karolinska Institutet, 171 77 Stockholm, Sweden; bengt.linderoth@gmail.com; 7Karolinska University Laboratories, Karolinska University Hospital, 184 50 Stockholm, Sweden; 8Theme Aging, Karolinska University Hospital, 141 86 Huddinge, Sweden

**Keywords:** Alzheimer’s disease (AD), astrocytes, amyloid beta (Aβ), encapsulated cell biodelivery (ECB), nerve growth factor (NGF), drug delivery strategy optimization, inflammation

## Abstract

Alzheimer’s disease (AD) treatment is constrained due to the inability of peripherally administered therapeutic molecules to cross the blood–brain barrier. Encapsulated cell biodelivery (ECB) devices, a tissue-targeted approach for local drug release, was previously optimized for human mature nerve growth factor (hmNGF) delivery in AD patients but was found to have reduced hmNGF release over time. To understand the reason behind reduced ECB efficacy, we exposed hmNGF-releasing cells (NGC0211) in vitro to human cerebrospinal fluid (CSF) obtained from Subjective Cognitive Impairment (SCI), Lewy Body Dementia (LBD), and AD patients. Subsequently, we exposed NGC0211 cells directly to AD-related factors like amyloid-β peptides (Aβ_40/42_) or activated astrocyte-conditioned medium (Aβ_40/42_/IL-1β/TNFα-treated) and evaluated biochemical stress markers, cell death indicators, cell proliferation marker (Ki67), and hmNGF release. We found that all patients’ CSF significantly reduced hmNGF release from NGC0211 cells in vitro. Aβ_40/42_, inflammatory molecules, and activated astrocytes significantly affected NGC0211 cell proliferation without altering hmNGF release or other parameters important for essential functions of the NGC0211 cells. Long-term constant cell proliferation within the ECB device is critically important to maintain a steady cell population needed for stable mNGF release. These data show hampered proliferation of NGC0211 cells, which may lead to a decline of the NGC0211 cell population in ECBs, thereby reducing hmNGF release. Our study highlights the need for future studies to strengthen ECB-mediated long-term drug delivery approaches.

## 1. Introduction 

Alzheimer’s disease (AD) is associated with early loss of basal forebrain cholinergic neurons (BFCNs) function, among other pathological features including amyloid-beta (Aβ) accumulation, astroglial activation, and inflammation [1,2]. BFCNs are highly dependent on the mature form of neurotrophin–nerve growth factor (mNGF) to maintain their survival and cholinergic phenotype during development and adulthood [1,3]. Dysregulation of the mNGF biosynthesis cascade has been reported in AD, predominantly leading to the accumulation of its precursor (proNGF) [4,5,6]. Concomitantly, a reduced ratio of NGF receptors, Tropomyosin receptor kinase A (TrkA—the high-affinity mNGF-specific receptor) to p75 neurotrophin receptor (p75NTR—low-affinity mNGF receptor), has been observed in the basal forebrain during AD progression [7,8,9], further hampering mNGF uptake and altered downstream signaling, leading to cognitive dysfunction [1,10]. 

Delivery of mNGF was envisioned to increase BFCN cell survival by reviving TrkA signaling present locally on cell bodies, which are still intact during the AD continuum [1]. This activation of BFCN TrkA signaling may stimulate acetylcholine production, activate cholinergic synaptic activity, re-establish cholinergic neurotransmission, and enhance innervation to the cortical and hippocampal regions crucial for cognition and memory function [11,12,13]. Until now, mNGF delivery in the AD patient’s brain has been accomplished by various methods including direct ventricular injection, gene therapy (using viral vectors), or encapsulated cell biodelivery (ECB) [14]. The ECB device is a hollow capsular device surrounded by a semi-permeable membrane (280 kDa cut-off), which harbor genetically modified cells growing in a 3-D matrix and can be retrieved from the brain after intended durations of therapy [15]. ECB implantation facilitates local delivery of therapeutic molecules over a long time, which needs an active cell population to be present inside ECB and had been used to deliver various therapeutic molecules in different conditions, such as epilepsy [14]. The precision of the delivery and tolerability of ECB-NGF therapy have been previously reported by our group in an open-label phase 1b clinical trial [16,17]. It was previously observed that human mature NGF (hmNGF) release from the ECB’s containing the genetically modified human retinal pigment epithelial (ARPE-19) cell line were affected when they remained implanted over time and assessed following explantation from human or animal brains [17,18]. 

It has been shown before in cases of age-related macular degeneration (AMD) that inflammatory molecules and Aβ peptides can impair retinal pigment epithelial (RPE) cells [19]. AD pathology is also associated with soluble inflammatory factors, cytokines, complement proteins, and Aβ peptides which may diffuse into the ECB owing to their small size (<280 kDa; ECB membrane molecular cut-off). Moreover, the surgical ECB implantation procedure may result in local tissue damage, glial activation, and capillary vessel disruption, leading to increased inflammatory conditions [20]. All these factors may potentially affect the cells present inside the ECBs, but due to technical limitations, these issues cannot be studied in vivo. Thus, using an in vitro experimental set-up, the effect of these factors on hmNGF-releasing cells needs to be investigated. 

Our previous studies using the first generation of hmNGF-releasing cells (termed NGC0295) showed that these cells were physiologically under stress and are sensitive to the inflammatory molecule interleukin-1beta (IL-1β), which affected hmNGF release over time [21]. To increase hmNGF release, a second generation of hmNGF-producing cells (termed NGC0211) was developed using transposon-mediated gene transfer, which releases ~10 times more hmNGF (∼400 ng hmNGF/10^−6^ cells/24 h) than NGC0295 cells (∼30 ng  hmNGF/10^−6^ cells/24 h) [18]. In our previous clinical trial study [17], when NGC0211 cell-containing ECBs were implanted within the human brain, we observed altered hmNGF release from the ECB devices as follows: 13 out of a total of 16 implants released some hmNGF whereas 3 implants failed to release detectable amounts of hmNGF. Among the devices that released hmNGF, eight implants released hmNGF at the same rate or higher than pre-implantation levels. To understand the reason behind this variable efficiency of hmNGF release from ECBs, we need to investigate whether the NGC0211 cell viability or activity is altered after exposure to CSF or other AD-associated molecules. In the present study, we assessed whether AD-associated factors like soluble Aβ peptides (Aβ_40/42_) may affect the hmNGF-releasing capacity of NGC0211 cells, (1) either directly by inducing stress and toxic effects, or (2) by activating astroglial cells to release inflammatory molecules, which, in turn, may affect NGC0211 cells. 

## 2. Materials and Methods

### 2.1. Plasmid Preparation and Generation of NGC0211 Cells

Preparation of the plasmid has been described elsewhere in detail [18]. Briefly, PCR-amplified HEK293 genomic DNA was cloned in pcDNA3.1(+) vector (Invitrogen, Gothenburg, Sweden), and modified to contain cytomegalovirus promoter/chimeric intron from pCI-neo (Promega, Madison, WI, USA) along with the cytomegalovirus early enhancer element/chicken beta-actin (CA) promoter sequence from pCAIB. From the resulting plasmid, the hmNGF and neomycin resistance cassettes were excised as a single fragment and inserted into the Sleeping Beauty (SB) substrate vector pT2BH, to generate the final plasmid termed pT2.CAn.hNGF.

The human RPE cell line, ARPE-19 (ATCC, Manassas, VA, USA), was cultured in DMEM/F12 medium supplemented with GlutaMAX (Invitrogen) and 10% fetal bovine serum (FBS) (Hyclone, Logan, UT, USA) at 37 °C and 5% CO_2_. ARPE-19 cells were co-transfected with pT2.CAn.hNGF and pCMV-SB-100X (expressing hyperactive SB transposase without antibiotic resistance element) plasmids using FuGENE (Roche, Basel, Switzerland) according to the manufacturer’s protocol. Transfected cells were selected using G418 (Sigma-Aldrich, St. Louis, MO, USA) and single-cell clonal colonies were expanded. The clone NGC0211 was used for the previous human phase 1b clinical trial [17], and therefore, was selected for use in the present study.

### 2.2. Two-Dimensional Cell Cultures

NGC0211 cells were cultured in complete DMEM/F12 medium containing GlutaMAX and 10% heat-inactivated FBS at 37 °C and 5% CO_2_. Confluent (85–90%) NGC0211 cultures were split until they were used for experiments (passage number 10–20). For splitting, cells were washed twice with phosphate-buffered saline (PBS) (Invitrogen), trypsinized using TrypLE Express Enzyme (Life Technologies, Carlsbad, CA, USA), and replated as a 1:3 ratio into T-75 flasks (Corning, New York, NY, USA) in culture medium.

For human CSF exposure (discussed in Section 2.10), NGC0211 cells were cultured in human endothelial serum-free medium (HE-SFM) (Invitrogen). The choice of medium was dictated by the necessity to avoid serum constituents from the culturing process and assess the specific contribution of CSF in modulating NGC0211 cells’ response. Cells were maintained in HE-SFM for 24 h before initiating CSF exposure. HE-SFM media was used since NGC0211 cells do not grow well in serum-free DMEM/F12 (unpublished data). 

Cortical human primary astrocytes (Cat no.1800) and all related culture medium supplements were purchased from ScienCell Research Laboratories (Carlsbad, CA, USA). Cells were initially grown and subsequently cultured on poly-L-lysine-coated T-75 flasks in astrocyte culture medium (Cat no. 1801) containing 2% FBS (Cat no. 0010), 1% Penicillin-Streptomycin concentrate (Cat no. 0503), and 1% of astrocyte growth supplement (Cat no. 1852), following the manufacturer’s recommendations. Sub-culturing was done when cells reached 80–85% confluency in a 1:4 ratio. All cell culture experiments were completed utilizing 2nd–5th passage number.

### 2.3. Aβ Peptide Preparation and Cell Culture Exposure

Aβ peptide (Aβ_40_ and Aβ_42_) soluble oligomers were prepared as reported earlier, with minor adjustments [22]. Briefly, vials of human Aβ_40_ or Aβ_42_ peptides expressed in E. coli (rPeptides, Lelystad, The Netherlands) were incubated for 30 min at room temperature (RT) and then dissolved in dimethyl sulfoxide (DMSO) to obtain stock concentrations of 0.5 mM, respectively. Resulting solutions were vortexed vigorously, sonicated at 40 Hz (Branson 2510 bath sonicator, Sigma-Aldrich, St. Louis, MO, USA) for 10 min, aliquoted, and stored at −20 °C until use. For the experiments, aliquots were thawed and diluted to working concentrations (1, 0.5, 0.1, 0.05 μM) in respective culture medium and used immediately for cell exposure. It has been shown previously that oligomeric Aβ peptides when diluted in culture medium maintain their oligomeric form at least until 70 h [23]. The FBS concentration in direct exposure is matched with that of the indirect exposure (explained in Section 2.4), maintaining ~5% FBS concentration during the exposure period. 

### 2.4. Astrocyte-Conditioned Media (ACM)

Following trypsinization, astrocytes were plated on poly-L-lysine-coated 24-well plates (7 × 10^4^ cells/well) and left for 24 h to recover. Medium was thereafter changed, and adhering cells were treated with either of the following: Aβ_40/42_ peptide (1, 0.5, 0.1, 0.05 μM), tumor necrosis factor alpha (TNFα, 20 ng/mL), or IL-1β (2 ng/mL) used here as a positive control for astrocyte activation or left unstimulated (untreated control) in a total volume of 500 μL. After 24 h of exposure, supernatant was collected and designated as astrocyte-conditioned medium (ACM). The ACMs were mixed with complete DMEM/F12 medium (50:50% media ratio) and used to stimulate NGC-0211 cells or ECBs. Equivalent amounts of TNFα and IL-1β were used to study their direct effect on NGC0211 cells following the same experimental set-up.

### 2.5. ECB Device Preparation and Treatment Exposure

Seven-millimeter-long ECB devices were manufactured from semi-permeable (280 kDa mean molecular weight cut-off) polysulfone hollow fiber membranes (Gloriana Therapeutics, Rhode Isalnd, RI, USA) threaded with a polyester terephthalate (PET) yarn matrix (Swicofil, Emmen, Switzerland). Each device was filled with 6 μL of cell suspension (10,000 cells/μL HE-SFM) by a semiautomatic custom-made cell injector system (Kineteks, Warwick, IL, USA) and sealed using a photopolymerized acrylic adhesive (Dymax, Torrington, CT, USA). Devices loaded with NGC0211 cells were maintained in 1 mL of HE-SFM medium at standard cell culture conditions for 2–3 weeks with weekly media replacements before initiating experimentation.

Prior to experimentation, the ECBs were incubated in complete DMEM/F12 media for 1 week. Initially, ECBs were cultured in 1 mL of DMEM/F12 complete medium for 4 h, and 500 μL of medium were collected as pre-exposure samples, which were kept in a −80 °C freezer until use. ECBs were then exposed to Aβ_40/42_ peptides (1 μM) or ACMs (astrocytes treated with 1 μM Aβ_40/42_) and incubated for 7 days. ECBs were then transferred to 1 mL of fresh complete DMEM/F12 medium, incubated for 4 h, and once more, 500 μL of medium were collected as post-exposure samples, and saved in a −80 °C freezer for future hmNGF ELISA analysis. 

In the remaining medium along with the ECBs, 50 μL of 10× alamarBlue (Invitrogen) were added to measure the total metabolic activity of the cells, mixed, and incubated for 1 h. From each well, 100 μL were drawn in triplicate, plated in a black bottom 96-well plate (Corning, New York, NY, USA), and fluorescence was read at 560 nm/590 nm (excitation/emission) in a spectrophotometer (Safire II Plate reader, Tecan, Männedorf, Switzerland) with a 5 nm bandpass filter and top read mode. 

### 2.6. Biochemical Measurements

To ascertain the ability of Aβ_40/42_ peptides or ACMs to induce cellular stress, biochemical measurements for the following parameters were performed: reactive oxygen species (ROS, 20 μM H_2_DCFDA, 485/520 nm), total glutathione (GSH, 50 μM mBCL, 394/490 nm), mitochondrial membrane potential (ΔΨm, 0.2 μM TMRM, 548/574 nm), as well as overall metabolic activity using alamarBlue (560/590 nm). All chemicals were obtained from Invitrogen. To perform these experiments, cells (1 × 10^4^ cells/well/100 μL) were plated onto 96-well clear bottom black plates (Corning, New York, NY, USA) and experiments were conducted for early (0–1 h kinetics, 3 h endpoint) and late (24 h endpoint) time points, respectively. Measurements were taken at a constant temperature of 37 °C using a spectrophotometer (top read mode to measure metabolic activity and bottom read mode for all other parameters).

For kinetic measurements up to 1 h, for ROS and GSH levels, cells were pre-incubated with H_2_DCFDA or mBCL for 20 min followed by Aβ_40/42_ peptides or ACMs exposure in a final volume of 100 μL/well. The plate was then returned to the incubator and finally 3-h end-point data were acquired from the same plate. To measure ΔΨm post 3 h of exposure, TMRM was added for the last 20 min followed by washing the cells twice with PBS and fluorescence was acquired in 100 μL of PBS. Similarly, to measure metabolic activity at 3 h, alamarBlue was added during the last 1 h and data was acquired. To evaluate late effects on biochemical parameters, endpoint readings were taken after 24 h of exposure to Aβ_40/42_ peptides or ACMs. For H_2_DCFDA, mBCL, or TMRM, probes were added for the last 20 min and the fluorescence measurements were recorded as mentioned above. For metabolic activity, alamarBlue was added 1 h before the recording of the activity measurement by the spectrophotometer as mentioned previously. 

### 2.7. Immunocytochemistry and Image Analysis

The proliferative ability of NGC0211 cells was evaluated by staining for the proliferation-associated protein Ki67. Briefly, cells (0.5 × 10^4^ cells/well/100 μL) were plated in a 16-well chamber slide (Lab-Tek, ThermoFisher Scientific, Gothenburg, Sweden) and exposed to various conditions. After 24/48/72 h of incubation, cells were washed 3× with PBS, fixed using phosphate-buffered formaldehyde 4% (*v/v*) (Sigma-Aldrich, St. Louis, MO, USA) for 5 min at RT, and permeabilized using 0.2% Triton X-100 (Sigma-Aldrich, St. Louis, MO, USA) for 15 min. Following 3× wash with PBS-T (PBS+ 0.05% Tween20), cells were then blocked using 1% bovine serum albumin (BSA, Sigma-Aldrich, St. Louis, MO, USA) for 30 min at RT, and incubated with primary mouse anti-human Ki67 antibody (1:100 dilution; clone MIB-1, DAKO, Glostrup, Denmark) in staining buffer (PBS-T + 1% BSA) overnight at 4 °C. Cells were then washed 3× with PBS-T, blocked (PBS-T + 1% BSA + 3% goat serum) for 15 min, and re-probed with conjugated Alexa-488 goat anti-mouse IgG secondary antibody (1:500 dilution, Invitrogen) for 2 h at RT. Cells were washed again 3 × with PBS-T, air dried, and mounted with DAPI containing mounting medium (VectaShield, Vector Laboratories Inc, Oxford shire, England). As a negative control, the cells were incubated only with goat anti-mouse Alexa-488 conjugated secondary antibody. Multiple images from different sample replicates were visualized and images were captured using an inverted laser scanning microscope (LSM 510 META; Zeiss, Germany). The Ki67 immunoreactivity was evaluated as the percentage of Ki67-positive cells out of the total population (Adobe Photoshop, San Jose, CA, USA), where at least 500 cells/well were counted. The cells were evaluated from three independent experiments.

### 2.8. Cell Death Assay 

To determine the percentage of viable, apoptotic, and necrotic cells following Aβ_40/42_ peptide or ACM exposure, treated NGC0211 cells were stained with FITC-Annexin-V (V) and propidium iodide (PI) (Invitrogen) according to the manufacturer’s protocol. Briefly, cells (7 × 10^4^ cells/well for 24/48/72 h or 3 × 10^4^ cells/well for 7 days) were seeded on 24-well plates (Corning, New York, NY, USA) and left in 500 μL of complete medium overnight. Cells were then exposed to Aβ_40_ or Aβ_42_ (final concentrations of 1, 0.5, 0.1, 0.05 μM) or ACMs and incubated for respective time points in a total volume of 500 μL. To have a complete picture of the total cells and the viability, the floating cells were collected after treatment with Aβ−peptides or ACMs and kept separately before the next step. Then, the attached NGC0211 cells were washed 3× with PBS, trypsinized, and collected by centrifugation (1500× *g*, 4 °C, 5 min) and pooled with the floating cells obtained in the previous step (total pool of treated cells = attached + floating) and resuspended in annexin binding buffer. Further, post-exposure supernatant was also separated from floating cells by centrifugation (3000× *g*, 4 °C, 5 min) and stored at −80°C for future hmNGF measurements.

Cells were then stained according to the manufacturer’s instructions, and data was acquired using a BD-Accuri C6 Plus flow-cytometer (BD Biosciences, Franklin Lakes, NJ, USA). The dual-color analysis allowed the identification of different cell populations: viable cells (Annexin V^−^PI^−^), early apoptotic cells (Annexin V^+^PI^−^)_,_ late apoptotic cells (Annexin V^+^PI^+^), and necrotic cells (Annexin V^−^PI^+^). 

### 2.9. Measurements of hmNGF by ELISA

The amount of hmNGF levels released by NGC0211 cells, astrocytes, or ECBs after treatment with Aβ_40/42_ peptides or ACMs was measured using a commercial ELISA kit (Cat No. DY256, R&D Systems, USA; assay range 31.2–2000 pg/mL), with minor modifications as described previously [24]. Briefly, 50 μL/well of capture antibody (2.0 μg/mL in carbonate buffer, pH 9.8) were plated in a 384-well plate (Nunc Maxisorp, Corning, Gothenburg, Sweden) and incubated overnight. Plates were then washed once for 5 min with 100 μL/well tris-buffered saline, blocked with 100 μL/well of 5% BSA in carbonate buffer for 1 h at RT, washed 3× with 100 μL/well TBS-T (TBS, 0.05% Tween 20, 0.01% NaN_3_), and incubated overnight with 50 μL/well supernatant samples or standards at 4 °C. The following day, plates were washed 3× with 100 μL/well TBS-T and incubated with 50 μL/well biotinylated detection antibody for 3 h at RT. Plates were then washed 3× with 100 μL/well TBS-T and incubated with 50 μL/well streptavidin-alkaline phosphatase (Streptavidin-AP, 1:10,000 dilution, Roche, Basel, Switzerland) for 1 h at RT. Finally, plates were washed 2× with 100 μL/well TBS-T followed by one washing with DEA buffer (1 M Diethanolamine buffer, pH 9.8). The alkaline phosphatase substrate (p-Nitrophenyl-Na_2_-6H_2_O, 1 mg/mL in DEA buffer) was then added 50 μL/well and colorimetric data were acquired kinetically in a pate reader (Safire II, Tecan) every 5 min for 1 h at 540 nm. Data were analyzed from standard curve plotted in reagent diluent (S1–S10, S1 = 2 ng/mL). For direct Aβ-treated NGC0211 cells, untreated groups were chosen as the control. When treated with astrocyte-conditioned medium, conditioned medium obtained from untreated astrocytes was set as the control. 

### 2.10. Human CSF Collection and Cell Exposure

Human CSF from AD (*n* = 17), Lewy body dementia (LBD) (*n* = 14), and subjective cognitive impairment (SCI) (*n* = 19) patients were obtained from the GEDOK database and biobank available at Karolinska University hospital memory clinic at Huddinge, Stockholm. These patients were diagnosed based on memory clinic evaluations including clinical examinations, cognitive tests, CSF analysis for Aβ and tau, and magnetic resonance imaging (MRI). The ethical permission for this study (Dnr: 2015/791-31/4) was obtained from the Regional Ethical Review Board of Stockholm.

NGC0211 cells (2 × 10^4^ cells/well) were plated in 96-well clear bottom black plates and allowed to grow for 24 h (as described in Section 2.2). The CSF to HE-SFM ratio was previously optimized [21], and the cells were thus treated with 200 μL/well, 50:50 mix of human CSF and HE-SFM medium. After 48 h of exposure, supernatant was collected and replaced with 200 μL/well fresh HE-SFM. Following another 4 h of incubation, 100 μL/well supernatant was collected and saved for future hmNGF analysis. In every well, 100 μL of HE-SFM media containing 2× alamarBlue were added and incubated for another 1 h after which fluorescence was read for alamarBlue as previously described. 

### 2.11. Statistical Analysis

Quantitative data are presented as mean ± standard error of the mean (SEM). Statistical analyses on the in vitro studies were performed by one-way ANOVA with a Tukey’s multiple comparison test for comparison of three or more groups or by two-way ANOVA as appropriate, followed by Tukey’s multiple comparison test. Statistical analyses were performed using Prism 8 software. Spearman correlation analyses were performed with SPSS software (version 2021). Results with * *p* < 0.05, ** *p* < 0.01, *** *p* < 0.001 were considered significant. 

## 3. Results

### 3.1. Release of hmNGF Is Altered by Exposure to Human Patient CSF (SCI, LBD, AD)

To ascertain the effect of human CSF on hmNGF release from NGC0211 cells, we exposed NGC0211 cells to human CSF for 48 h (AD = 17, SCI = 19, LBD = 14). The ratio between total tau versus Aβ_42_ (t-Tau/Aβ_42_) obtained from the CSF data was plotted for disease type segregation (Figure 1A). Exposure of NGC0211 cells to patients’ CSF (SCI, LBD, and AD) resulted in significant decreases in hmNGF release when compared to the HE-SFM medium as a control (*p* < 0.001) (Figure 1B). To evaluate whether this effect was due to change in cellular activity or viability, we checked the overall metabolic activity of the cells, which remained unaltered (Figure 1C).

To ascertain whether the observed reduction in hmNGF release from NGC0211 cells was due to AD-specific factors, we performed Spearman’s correlation between hmNGF release from NGC0211 cells and the Aβ_42_ content of CSF. We observed a non-significant correlation between hmNGF release from NGC0211 cells and CSF Aβ_42_ content (Figure 1D,E), without affecting the metabolic activity of the cells. Nevertheless, since the levels of Aβ in the CSFs were in low nanogram levels (average values: SCI = 955 ± 344.043 pg/mL; LBD = 645 ± 232.412 pg/mL; AD = 572.444 ± 234.29 pg/mL), which did not affect hmNGF release from NGC0211 cells in vitro (Section 3.2.), the impact of CSFs on hmNGF release from NGC0211 cells may be due to some other factors associated with AD severity, which need further analysis and understanding.

### 3.2. Direct Exposure of Aβ_40/42_ Peptides Marginally Affected Cell Death, hmNGF Release, and Stress Response

Flow cytometry analysis demonstrated that Aβ_42_ exerted its toxicity at 24 h, showing higher early apoptotic populations (V^+^PI^−^) (~8%) than when treated with Aβ_40_ (~3%), and as compared with their respective controls (3.9 and 2.5%, respectively). However, when NGC0211 cells were incubated for 7 days, the species-specific and dose-specific effects on the V^+^PI^−^ cell population disappeared (Figure 2A,B). A time-dependent effect of Aβ_40_ exposure was observed on the accumulation of necrotic cells (V^−^PI^+^ cells), which was statistically significant after 7 days (~12% vs. 6.4% of control) (Figure 2C). This response was not noticed in NGC0211 cells exposed to Aβ_42_ (~6% vs 7.08 % of control) (Figure 2D). Similarly, when late apoptotic cells were assessed (V^+^PI^+^), a time-dependent effect was observed for both Aβ_40_ and Aβ_42_ peptides (Appendix A). Taken together, these results suggest an early dose-dependent toxicity of Aβ_42_ and a late toxic effect of Aβ_40_ on NGC-0211 cells.

Culture supernatant from the same experiment as above was analyzed for hmNGF release from NGC0211 cells when exposed to Aβ peptides. An overall Aβ species-specific trend was observed at early time points, where Aβ_40_ primarily reduced hmNGF release while Aβ_42_ increased hmNGF release after 24 and 48 h, when compared to their respective controls. However, gradually, Aβ exposure resulted in a mild to significant increase in hmNGF release at later time points (72 h and 7 days) when compared to the control group (Figure 2E–H). Interestingly, the 0.5 μM concentration of both Aβ peptides showed reduced hmNGF release at early time points whereas the higher dose of 1 μM had the opposite impact, but this difference disappeared with time. 

Since cellular stressors (like Aβ peptides) may not induce cell death but can alter the functional properties of the cells, we evaluated various biochemical stress parameters at early (≤3 h) or late (24 h) timepoints. At 3 h (Figure 2I,J), when compared to their respective controls, Aβ_40/42_ peptide exposure resulted in reduced ROS levels without altering GSH levels. Fluctuations were observed in cellular metabolic activities, where Aβ_40_ caused a significant dose-dependent reduction, whereas Aβ_42_ exposure was significant with only the highest dose (1 μM, *p* < 0.05) (Figure 2K). However, mitochondrial activity as measured by membrane potential (ΔΨm) was found to be unaltered (Figure 2L).

We did not observe any significant impact on the stress parameters post 24 h exposure except that 1 μM Aβ_40_ induced minor changes in ΔΨm depolarization at 24 h only (*p* < 0.05), when compared to the respective control (Appendix A). We also evaluated the cellular mitochondrial network as a stress marker [25] (Appendix A), which showed early disruption by Aβ_40_ at 24 h but subsided during the longer exposure time (48/72 h, respectively), especially when using 0.1 μM Aβ_40_. On the other hand, Aβ_42_ peptides seem to have lesser effects at any time point, when compared to Aβ_40_. Overall, our data shows that NGC0211 cells are less affected when exposed to Aβ peptides and can maintain their hmNGF release over a 7-day period.

### 3.3. Severe Anti-Proliferative Impact of Aβ_40/42_ Peptides on NGC0211 Cells

To understand whether Aβ peptides affect the NGC0211 cell proliferation rate, we exposed cells to various concentrations of Aβ_40/42_ (1, 0.5, 0.1, 0.05 μM), and evaluated Ki67 protein expression (as a marker for proliferation). We observed significantly reduced immunoreactivity for Ki67 indicating hampered proliferation at all time points and doses of Aβ peptides (Figure 3), when compared to their respective control groups. Although the anti-proliferative impact was similar at 24 h, dose-dependent effects appeared at later time points (72 h). Among the Aβ peptides, Aβ_42_ showed marginally stronger anti-proliferative effects at all time points when compared to the respective control. Overall, we found that both Aβ peptides have a severe anti-proliferative effect on NGC0211 cells, and this effect increases time dependently.

### 3.4. Astrocyte-Conditioned Media (ACM) Shows Minimal Impact on NGC0211 Cell Death, hmNGF Release, and Stress Response

Before initiating experiments using human primary cortical astroglial cells, we performed immunostaining with anti-S100 and anti-GFAP to ascertain their purity and activation status of astroglial cells (Appendix A). ACMs derived from Aβ-treated astrocytes are denoted as Aβ_40ACM_ and Aβ_42ACM_, respectively. Flowcytometric analysis of NGC0211 cells exposed to ACM for 24 h showed significant cytoprotective effects from Aβ_40ACM_, whereas low doses of Aβ_42ACM_ increased cell death, when compared to respective controls (Figure 4A,B). To check whether different species of Aβ differentially activate astrocytes, we performed complement-3 protein analysis from the ACMs but did not observe any difference, when compared to the control group (Appendix A). 

Simultaneously, we measured NGF released by the astrocytes after exposure to Aβ_40/42_, IL-1β, or TNFα for 24 h, and found differential effects from respective treatments (Appendix A). However, when NGC0211 cells were treated with Aβ_40ACM_, Aβ_42ACM_, and IL-1β/TNFα_ACM_, no significant effect of ACMs was observed on the hmNGF-releasing ability of NGC0211 cells, when compared to the control group (Figure 4C). To account for the effect of individual media types, astrocyte medium and DMEM/F12, on hmNGF release from NGC0211 cells, we incubated identical wells with fresh media. The addition of fresh astroglial media to NGC0211 cells did not have any significant effect on hmNGF release. To account for the long-term effects of ACM exposure, treatments (media controls, Aβ_40ACM_, Aβ_42ACM_, IL-1β/TNFα_ACM_) were replaced with fresh DMEM/F12 and hmNGF release was measured after further incubation for 24 h (total exposure for every well became 24 h + 24 h). Apart from a significant increase in the IL-1β_ACM_ group, hmNGF release was not significantly altered in other groups when compared to the untreated astrocyte control group (Appendix A).

We also measured the effect of ACMs on stress response parameters in NGC0211 cells. ACMs collected from untreated astrocytes showed significant alteration in various biochemical parameters when compared to NGC0211 cells grown in DMEM/F12 media at the 3 (Figure 4D–G) or 24 h (Appendix A) time points. However, the treated astrocyte ACMs (Aβ_40ACM_, Aβ_42ACM_, IL-1β/TNFα_ACM_) did not show any significant modulation of biochemical parameters in NGC0211 cells, when compared to untreated astrocytes, which served as an experimental control to observe any treatment-specific changes from astrocytes. All ACMs were found to alter the mitochondrial network after 3 h of exposure (Appendix A). These data imply that the cellular factors released by activated astrocytes in ACMs do not induce potential stress in NGC0211 cells, which could significantly alter them functionally, when compared to untreated astrocytes.

### 3.5. ACM’s Showed Significant Anti-Proliferative Effects on NGC0211 Cells

To understand whether the positive controls, IL-1β and TNFα, had any impact on proliferation, we exposed NGC0211 cells directly to IL-1β (2 ng/mL) or TNFα (20 ng/mL) (Appendix A). As previously observed with direct exposure to Aβ_40/42_ peptides, significant anti-proliferative effects were observed from all the treated ACM tested, when compared to respective controls (Figure 5). Aβ_40ACM_ showed stronger anti-proliferative effects at 24 and 48 h, but IL-1β/TNFα_ACM_ displayed more severe effects on Ki67 expression than Aβ peptides (Appendix A). The effect of Aβ_40ACM_ and Aβ_42ACM_ showed reduced Ki67 expression in a time-dependent manner until 48 h, after which the effect of Aβ_42ACM_ continued to deteriorate until 72 h, whereas Aβ_40ACM_ showed reduced efficacy, when compared to controls. The influence of Aβ_40ACM_ was dose dependent (except at 48 h) whereas Aβ_42ACM_ showed dose dependency only at 72 h, possibly indicating a long-term effect of Aβ_42_. When the anti-proliferative ability of different ACM was compared, IL-1β/TNFα_ACM_ induced more severe effects on NGC0211, among which TNFα_ACM_ displayed a significant long-term suppressive influence. These data show that activated astrocytes have an anti-proliferative effect on NGC0211 cells by reducing Ki67 protein expression.

### 3.6. Impact of Different Exposure Regimes on the ECB-NGF Device

Next, we sought to ascertain the effect of Aβ peptides and Aβ_ACM_ on NGC0211 cells growing inside the ECB device. Assessments of metabolic activity and hmNGF release were performed after 7 days of continuous exposure to either Aβ_40/42_ peptides directly or to the ACM (Aβ_40ACM_, Aβ_42ACM_) (Figure 6). We observed differential response from the ECB devices, wherein direct exposure to Aβ peptides reduced hmNGF release whereas exposure to Aβ_40ACM_ and Aβ_42ACM_ significantly increased hmNGF release, as compared to their respective controls (Figure 6A). Upon measurement of metabolic activity from the ECBs, we observed a difference between the exposure types (directly exposed versus ACM-exposed routines), but no alterations were observed when treatments were compared to their respective controls (Figure 6B). The difference in metabolic activity between the different treatment types was not observed in NGC0211 cells under the 2-D-culture set-up (Figure 4C), which may indicate a modified response of NGC0211 cells when grown in a 3-D support. We also did not observe any difference when different ECB devices harboring un-transfected ARPE-19, NGC0295, or NGC0211 cells were compared to each other (Appendix A). This data shows that Aβ_40/42_ peptides or the ACM (Aβ_40ACM_, Aβ_42ACM_) did not hamper hmNGF release and metabolic activity of the cells present within the ECB devices, as compared to the respective control groups.

## 4. Discussion

This study identified the significant anti-proliferative potential of Aβ_40/42_ peptides, inflammatory molecules (TNFα, IL-1β), and activated astrocytes on NGC0211 cells. These mechanisms could be one of the plausible reasons for the unpredictable hmNGF release from ECBs over time when implanted in AD patients [16,17]. We also report that Aβ_40/42_ peptides, inflammatory molecules, and activated astroglia do not significantly affect hmNGF release or survival of NGC0211 cells directly. Due to the lack of an effect of these factors on hmNGF release from NGC0211 cells, the acute influence of human CSF on hmNGF release might originate from other proteins not investigated in this study but are present in the diseased condition. Moreover, the effect of CSF is independent of cell death induction and needs further exploration. Conclusions drawn in this study can be attributed to all encapsulated cell-mediated drug delivery strategies since Aβ peptides and inflammatory molecules are present in high concentrations in the AD brain, whereas inflammation itself is a common part of several pathologies.

In the present study, we used soluble Aβ oligomers since increasing evidence indicates that soluble Aβ oligomers have high pathogenicity [26]. Due to their small size and increased concentration in AD brain tissue (up to 1.3 µM) [27], Aβ peptides can potentially diffuse inside the ECB device and induce NGC0211 cell dysfunction. Similar findings have been previously reported in AMD conditions, where Aβ peptides were shown to affect RPE cells [19]. Moreover, Aβ peptides have been reported to be present in high concentrations in mitochondria [28,29], where they can regulate mitochondrial bio-energetics, leading to ROS generation and hampered mitochondrial activity [30]. Although Aβ peptides are incapable of inducing cell death at a <5 μM concentration [21,31] (Figure 2), they can significantly induce inflammatory responses from ARPE cells, leading to degeneration and accelerated senescence [32,33]. Surprisingly, Aβ peptides were not found to induce oxidative stress in NGC0211 cells, which may be attributed to the antioxidant properties of soluble Aβ [34,35]. Nonetheless, we showed that Aβ peptides and inflammatory molecules, such as IL-1β and TNFα, can severely impair NGC0211 cell proliferation when exposed directly (Figure 3, Appendix A). Our data indicate that NGC0211 cells have the capability to adapt to Aβ_40/42_ peptide-induced stress with an extended exposure duration of up to 7 days (Appendix A).

Astrogliosis is also a common pathological feature in AD, where Aβ peptides themselves are also known to induce astrocytic activation, leading to the release of pro-inflammatory cytokines [36,37,38]. Astroglia play an important role in RPE cell maintenance, but under pathologically activated conditions, they disturb RPE activity and viability [39]. Specifically, altered inflammatory cytokine expression and increased interleukin-1 receptor antagonist (IL-1ra) expression from human RPE cells have been shown [40]. Previously, we reported the protective response of anti-IL-1Ra antibodies on first-generation NGC0295 cells [21]. Our current data shows that based on the hmNGF-releasing capacity and stress response (Figure 4), the second-generation NGC0211 cells are more resistant to IL-1β. Our data showed the inability of activated astrocytes to induce redox imbalance, defined as elevated ROS levels with concomitant GSH depletion, but showed effects on mitochondrial activity and connectivity (Figure 4 and Appendix A). Aβ peptides, IL-1β, and TNFα induced astrocyte-activation-mediated influence on NGC0211 cell survival, stress response, and hmNGF release, and were also found to be insignificant (Figure 4, Figure 5 and Figure 6). Although recent evidence shows the involvement of immune cells’ effects on encapsulated cells in an in vitro setting [41], the availability and accumulation of these cells might be comparatively less in immune-privileged organs like the brain [42], where glial cells are the major in situ contributors to inflammatory mediators. Nevertheless, due to the role of microglial cells, brain-resident macrophages, in initiating inflammation in the brain tissue, future studies are needed to understand whether microglial cells play a role in hampering encapsulated cells. Similarly, it would be interesting to evaluate the impact of tau and phosphorylated-tau species on NGC0211 cell function due to their increased presence in the brain tissue during the AD continuum.

Gradual accumulation in dead cells over time (Figure 2C,D) could be due to increased cell death or a gradual dysregulation of dead cell clearance pathways. However, we did not observe a considerable amount of cell death among NGC0211 cells following Aβ or ACM exposure (Figure 2A,B). RPE cells are capable of clearing dead cells by phagocytosis under physiological conditions [43,44], but whether Aβ peptides or inflammatory molecules can affect RPE phagocytotic activity in vitro is less understood. In the ECB setting, an inability to achieve effective dead cell clearance may lead to a build-up of dead cell bodies, which can induce stress [45]. Cell lines grown in ECB devices or culture dishes for a long time (confined space) may develop increased stress, leading to untimely death, and cellular replacement achieved by compensatory proliferation is needed to maintain a steady population of cells [46]. Hampered compensatory replication will lead to subsequent depletion of the overall cell population over time due to an accelerated rate of senescence (and slow proliferation). Our data shows that Aβ peptides, inflammatory molecules, and astroglial activation have a profound effect on the cell proliferation capacity of NGC0211 cells, suggesting their reduced ability to perform compensatory re-population activities (Figure 3, Figure 5 and Appendix A). This may have a profound influence on the long-term therapeutic application of ECB devices, since over time, depletion of cells may lead to a reduced release of therapeutic molecules. In the present study, hmNGF release data (Figure 2E–H) show that the acute effect of Aβ disappears with time, perhaps due to the metabolism of Aβ peptides within the culture set-up, but chronic exposure to Aβ combined with inflammatory molecules may show different effects.

Apart from a profound impact on NGC0211 cell proliferation, neither Aβ_40/42_ nor ACM were found to affect hmNGF release when cells were grown in conventional 2-D culture (Figure 2, Figure 4 and Appendix A). Interestingly, significantly higher amounts of hmNGF release were observed when NGC0211 cells growing in 3D support inside ECBs were incubated with ACM but not with Aβ_40/42_ direct exposure, without altering metabolic activity in any groups (Figure 6). For the direct exposure, a similar reduction of hmNGF release was also observed in the control ECBs, which were not exposed to any treatments, indicating that this response could be a treatment-independent effect. Contrarily, an increase in hmNGF release after Aβ_ACM_ exposure may not be due to the media composition (1:1 ratio; astrocyte: DMEM/F12) since we did not observe such an effect in previous 2-D cell culture experiments (Figure 4C and Appendix A). The increase may also not be due to factors exclusively released from ‘activated’ astrocytes since untreated astrocytes (control ECBs) showed a similar hmNGF release when compared to Aβ_ACM_-exposed ECBs. The increased hmNGF release post Aβ_ACM_ exposure is also independent of Aβ peptides since there is a contrasting observation from the direct Aβ exposure shown in the same figure (direct vs. indirect exposure). One probable factor could be the favorable change in NGC0211 cell behavior when grown in 3-D matrix, which may have enhanced its response to the factors present in the astrocyte-conditioned medium, which enhances hmNGF release or accelerates transgene transcription/translation from NGC0211 cells. We also found that the transfection method utilized did not affect the outcome of these exposures, and the outcome on metabolic activity was similar to the un-transfected ARPE-19 when grown in an ECB (Appendix A). These observations under the current experimental conditions may indicate the resilience of the NGC0211 cells towards activated astrocyte-mediated negative alteration to its hmNGF-releasing capability. Although astrocytes themselves have been reported previously as a major source of NGF [47], these levels do not significantly hamper our study analysis since astrocytic levels are much lower than the levels released from NGC0211 cells or ECB devices (Appendix A). A similar observation was found upon exposing NGC0211 cells to human CSFs, where hmNGF release was significantly reduced (Figure 1B) without considerable alteration of the metabolic activity (Figure 1C). As evident, the AD CSF did not affect metabolic activity in the NGC0211 cells, but a direct correlation was evident between Aβ_42_ levels and hmNGF release (Figure 1E). Since, in AD, the Aβ_42_ content in CSF is gradually decreased as the disease progresses, the data shows that hmNGF release from NGC0211 cells is decreased with increasing severity of AD pathology. 

Apart from various other methods of local drug delivery in the brain tissue [14], the ECB platform holds great promise in quantitative and precision drug delivery, which can be safely applied and removed using minimally invasive surgical procedures [15]. Apart from delivering hmNGF, the ECB platform has been successfully tested to deliver various class of proteins in different experimental settings [48,49,50,51]. Along with the prevailing inflammatory conditions within the degenerative/damaged brain, ECB implantation procedures may result in increased local inflammation due to glial activation and damage to blood vessels and capillaries. Simultaneously, following ECB implantation, it must be bathed with interstitial fluid/CSF within the brain tissue for optimal therapeutic function (outflow of hmNGF) and NGC0211 cell survival (inflow of nutrients and oxygen). Thus, the CSF composition and local presence of activated glia will play an important role in modulating ECB performance post implantation (especially molecules <280 kDa). As depicted in Figure 1, hmNGF release from NGC0211 cells was reduced under the influence of AD CSF and was correlated with Aβ_42_ levels. Since Aβ peptides were previously reported to affect RPE survival in AMD [33,52] and induce inflammation from astroglial cells [53,54], we assessed their effect on NGC0211 cell survival under 2-D and 3-D cell culture condition (Figure 2, Figure 4 and Figure 6). Interestingly, we failed to observe any significant influence of these conditions (Aβ exposure and astrocyte-mediated inflammation) on NGC0211 survival and hmNGF release, indicating that these cells can sustain their functions despite the presence of these stressors. However, the effect of Aβ, inflammation, and astroglial activation on cell proliferation will affect the maintenance of cell numbers inside the ECB devices, which eventually may compromise its long-term therapeutic efficacy. Moreover, the identification of pathways that mediate the anti-proliferative impact of Aβ, inflammation, and astroglial activation on NGC0211 cells may aid in the production of resistant cells which may advance the therapeutic life term of implanted ECBs. 

## 5. Limitations 

The major strength of this study is the use of relevant cell culture models, which are directly related to the encapsulated cell therapy targeted previously in AD patients. We also used primary cortical astrocytes from human postmortem brain tissue, thereby trying to recapitulate an appropriate model for astrocyte activation. Another strength of this study is the use of the 2-D and 3-D culture system, showing the differential response of encapsulated cells (NGC0211 cells). A primary limitation of this study is the use of an in vitro cell culture system instead of an animal model, but the reason for performing in vitro culture studies was to understand the specific contribution of Aβ and astrocyte activation towards dysregulation of NGC0211 cells. Animal model studies are often complicated by the fact that several factors are at play in affecting the ECB devices; nevertheless, in vivo studies are the next step to optimize ECB-NGF devices. Another limitation is the observed differential response of NGC0211 cells under the 2-D and 3-D culture set-up, which needs further investigation.

## 6. Conclusions 

We showed that NGC0211 cells are physiologically capable of resisting Aβ_40/42_ peptides and astroglial activation-induced stress and continue producing hmNGF in an acute setting. However, we found a significant anti-proliferative impact of Aβ_40/42_ peptides, inflammatory molecules, and astroglial activation, which may affect long-term maintenance of the NGC0211 population within the ECB device. Further development of the NGC0211 cells is warranted as a drug delivery vehicle to sustain long-term viability and efficacy in a clinical setting for AD therapy. Several other cell-based therapies have been targeted in AD, and our study may help explain the time-dependent reduction in their efficacy. 

## Figures and Tables

**Figure 1 cells-10-02834-f001:**
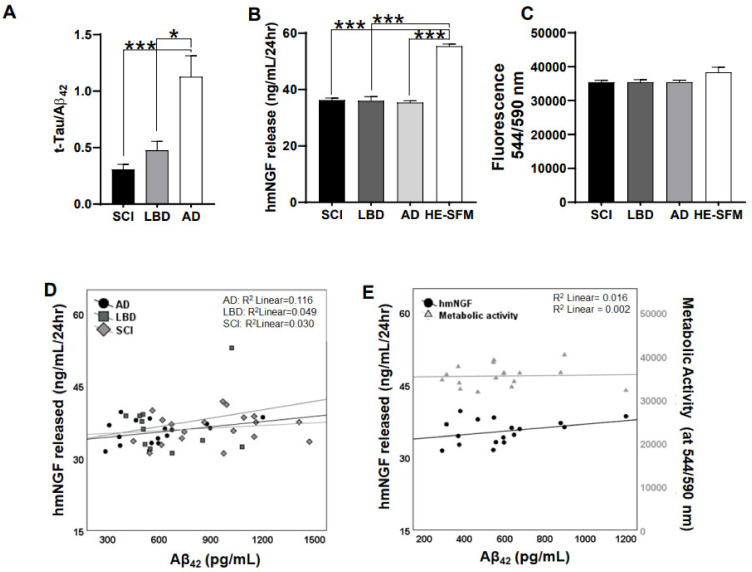
Effect of human CSF on NGC0211 cells. NGC0211 cells (1 × 10^4^ cells/well, 96-well plate) were cultured in vitro and exposed to human CSF (AD, SCI, LBD) for 48 h and parameters; hmNGF release and metabolic activity were measured in triplicates. (**A**) Absolute values of t-Tau and Aβ_42_ levels were obtained from the GEDOK database and plotted as ratios of t-Tau/Aβ_42_ (AD = 20, SCI = 20, LBD = 11) to ascertain pathological segregation of CSF samples. (**B**) After exposing NGC0211 cells to CSF for 48 h (AD = 17, SCI = 19, LBD = 14), cells were further incubated with fresh HE-SFM medium (see M&M) for 4 h and supernatant was collected to estimate the amount of released hmNGF by ELISA. (**C**) Using the same experiment (as Figure 1B), after taking out the cell supernatant, adhered cells were incubated with media containing 1 × alamarBlue for 1 h, followed by collection of supernatants to measure alamarBlue fluorescence (544/590 nm). (**D**) Scatter plots demonstrate the correlation between the release of hmNGF from NGC0211 cells and Aβ_42_ levels in CSF in the samples tested (AD, SCI, LBD). (**E**) Dual axis scatter plot demonstrating the correlation between the released hmNGF and metabolic activity of NGC0211 cells to the CSF-Aβ_42_ levels, though only within the AD group. Data are represented as mean ± S.E. Statistical analysis using one-way ANOVA analyses with a Tukey’s multiple comparison test was performed to compare control and treated groups in 1A–1C. Spearman correlation analyses was done in 1D and 1E. * *p* < 0.05, *** *p* < 0.001.

**Figure 2 cells-10-02834-f002:**
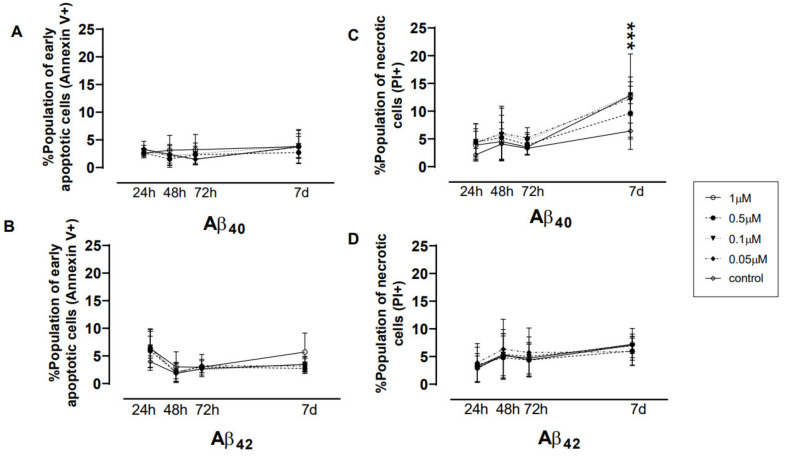
Impact of Aβ peptides on cell death, hmNGF release, and stress pathways: NGC0211 cells (7 × 10^4^ cells/well for 24/48/72 h or 3 × 10^4^ cells/well for 7 days) were seeded on 24-well plates and incubated with different Aβ_40/42_ concentrations (1, 0.5, 0.1, 0.05 μM) for different time points (24, 48, 72 h, and 7 days) (**A**–**H**). Supernatant was collected for measurement of hmNGF by ELISA and the cells were further trypsinized and stained with Annexin-V/propidium Iodide (PI) dyes. (**A**–**D**) Cells stained with Annexin-V/PI dyes were acquired in a flow cytometer and data is represented as the stained percent population as follows: early apoptotic cell death (Annexin V^+^), and dead/necrotic cells (Annexin PI^+^). (**E**–**H**) The culture supernatant post-Aβ exposure was analyzed for hmNGF content using ELISA and data is represented for different time points. (**J–L**) NGC0211 cells were treated with various Aβ_40/42_ concentrations (1, 0.5, 0.1, 0.05 μM) and evaluated up to 3 h for ROS production (**I**), GSH content (**J**), metabolic activity (**K**), and mitochondrial membrane potential (ΔΨm) (**L**) to study the immediate impact on stress-related pathways. Data are represented as mean ± S.E. (*n* = 3). Statistical analysis using one-way ANOVA analyses with a Tukey’s multiple comparison test was performed to compare control and treated groups. * *p* < 0.05, ** *p* < 0.01, and *** *p* < 0.001.

**Figure 3 cells-10-02834-f003:**
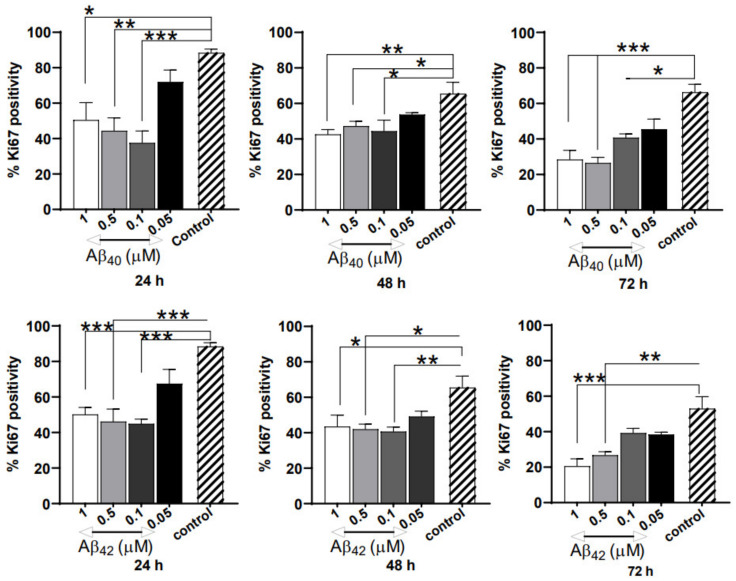
Dysregulation of NGC0211 proliferation by Aβ peptides. To study whether cellular proliferation is affected, NGC0211 cells (0.5 × 10^4^ cells/well) were plated in a 16-well chamber slide and exposed to various Aβ_40/42_ concentrations (1, 0.5, 0.1, 0.05 μM) for different time points (24, 48, 72 h). Post exposure, cells were fixed and probed with mouse anti-human Ki67 antibody (1:100 dilution; clone MIB-1, DAKO, Glostrup, Denmark) followed by Alexa-488-anti mouse antibody (1:500 dilution, Invitrogen). Slides were mounted and imaged using an inverted laser scanning microscope. At least 3 images from individual samples were counted for Ki67 immunoreactivity and data is presented as Ki67-positive cells taken as a percentage of the total cell population (DAPI staining). Data are represented as mean ± S.E. (*n* = 3). Statistical analysis using one-way ANOVA analyses with a Tukey’s multiple comparison test was performed to compare control and treated groups. * *p* < 0.05, ** *p* < 0.01, and *** *p* < 0.001.

**Figure 4 cells-10-02834-f004:**
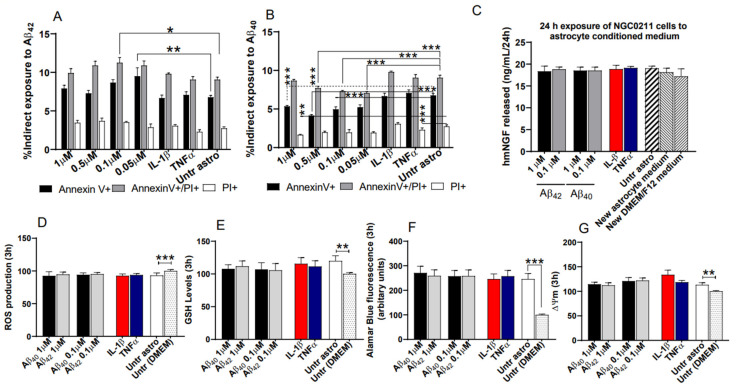
Activated astrocyte-conditioned medium (ACM) modulates NGC0211 cell survival, hmNGF release, and stress pathways. Astrocytes (7 × 10^4^ cells/well) were cultured in 24-well plates and challenged with Aβ_40/42_ peptides (1, 0.5, 0.1, 0.05 μM), inflammatory molecules (TNF-α 20 ng/mL, IL-1β 2 ng/mL), or left untreated (control) for 24 h. The astrocyte condition medium (ACM) supernatant was collected, diluted equally with complete DMEM/F12 medium, and used to treat NGC0211 cells for 24 h. (**A**,**B**) Treated NGC0211 cells were harvested, stained with Annexin-V/PI dyes, and acquired in a flow cytometer. Data represent the percentage of the stained population as follows: early apoptotic cell death (Annexin V^+^), late apoptotic cell death (Annexin V^+^PI^+^), and dead/necrotic cell (Annexin V^-^PI^+^). Statistical analysis using two-way ANOVA analyses with a Tukey’s multiple comparison test was performed. * *p* < 0.05, ** *p* < 0.01, and *** *p* < 0.001. (**C**) Following NGC0211 cell treatment with ACMs for 24 h, the culture supernatant was analyzed for released hmNGF content using ELISA. (**D**–**G**) Similarly, after treatment of NGC0211 cells for 3 h with ACM, cells were probed with different dyes to assess early stress factors like ROS (D), GSH (**E**), metabolic activity (**F**), and mitochondrial membrane potential (ΔΨm) (**G**). Data are represented as mean ± S.E. (*n* = 3). Statistical analysis using one-way ANOVA analyses with a Tukey’s multiple comparison test was performed to compare control and treated groups. * *p* < 0.05, ** *p* < 0.01, and *** *p* < 0.001.

**Figure 5 cells-10-02834-f005:**
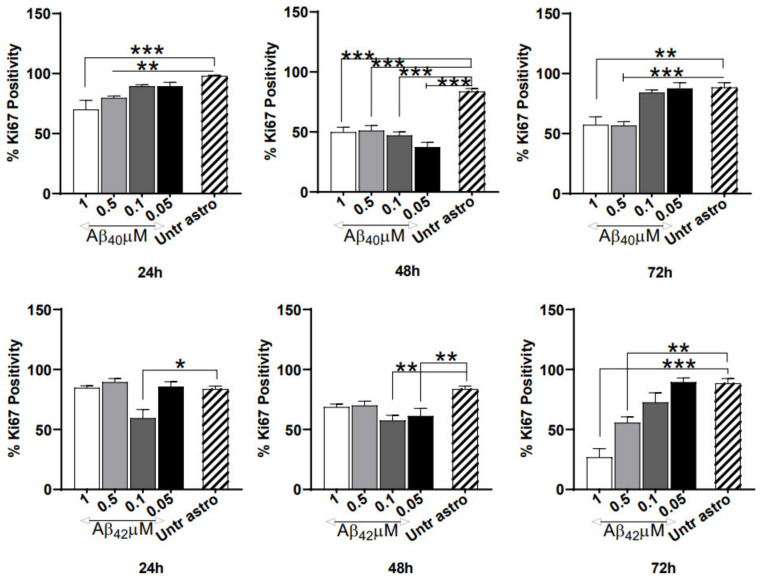
ACM displays an anti-proliferative effect on NGC0211 cells. Astrocyte-conditioned medium (ACM) was obtained after treating astrocytes (7 × 10^4^ cells/well, 24 well plates) with Aβ_40/42_ peptides (1, 0.5, 0.1, 0.05 μM), inflammatory molecules (TNFα 20 ng/mL, IL1-β 2 ng/mL), or left untreated (control) for 24 h. ACM was used to treat NGC0211 cells (0.5 × 10^4^ cells/well, 16-well chamber slide) for 24 h. Cells were then fixed, stained with mouse anti-human Ki67 antibody (1:100 dilution; clone MIB-1, DAKO, Denmark) followed by incubation with secondary Alexa-488-conjugated anti-mouse antibody (1:500 dilution, Invitrogen). The slides were mounted using DAPI containing mounting medium. Slides were then imaged, and individually stained cells were counted. Data are represented as mean ± S.E (*n* = 3). The quantification of Ki67-positive cells was based on the evaluation of at least 500 cells/well. Statistical analysis using one-way ANOVA analyses and additional Tukey’s Multiple Comparison Test was performed to compare control and treated groups. * *p* < 0.05, ** *p* < 0.01, and *** *p* < 0.001.

**Figure 6 cells-10-02834-f006:**
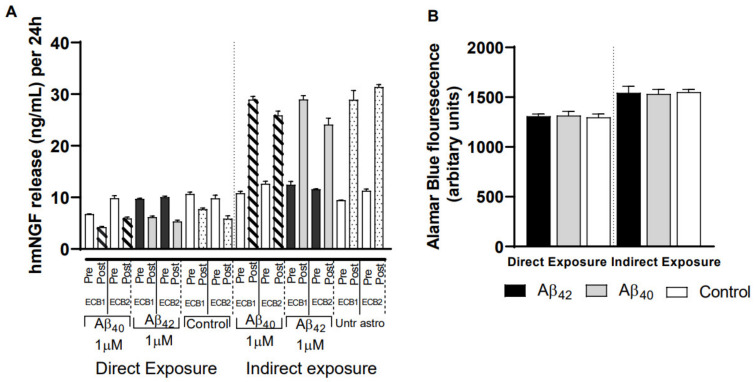
Aβ_40/42_ peptides and ACM differentially regulate hmNGF release from ECB devices. ECB devices were maintained in DMEM/F12 media for few weeks before exposing them to Aβ_40/42_ peptides (1 μM) or ACM (treated with 1 μM Aβ_40/42_ peptide) for 7 days in a final volume of 1 mL. (**A**) Following 7 days of exposure, media was replaced with 1 mL of fresh DMEM/F12, and ECB devices were further incubated for 4 h. The amount of hmNGF released during this 4 h was measured using ELISA (R & D Systems) by withdrawing 500 µL of media. The values obtained were multiplied by a factor of 6 to convert data to hmNGF released over 24 h. Data from individual ECB devices are represented. (**B**) To measure the impact of the exposure on metabolic activity, 1 × alamarBlue was added and incubated for 1 h. In total, 100 µL of medium were drawn in triplicates and fluorescence was measured at 540/590 nm in a Tecan plate reader using a black bottom 96-well plate. Data are represented as mean ± S.E. (*n* = 2).

## Data Availability

The datasets used and/or analyzed during the current study are available from the corresponding author on reasonable request.

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
