# Peer review of "Amyloid-Beta Peptides and Activated Astroglia Impairs Proliferation of Nerve Growth Factor Releasing Cells In Vitro: Implication for Encapsulated Cell Biodelivery-Mediated AD Therapy"

_cells, 2021, doi:10.3390/cells10112834_

Round 1

Reviewer 1 Report

In this manuscript, Mitra and colleagues aimed at unveiling the effects of some molecules on the long-term survival/proliferation/NGF delivery capability of ECB devices composed of encapsulated NGC0211 cells. While I can found the aim of the manuscript interesting, I have to enlighten that most of the analyses here reported have been made on cells growing as monolayer in a condition that has limited relevance to the ECB device (3D growth). Notably, analyses performed on proper growth conditions (ECB, i.e. 3D conditions) have shown no major differences following the treatments considered, showing major difference with monolayer growth.

Overall, this aspect limits the enthusiasm on this study.

Additional comments:

  • Authors should revise experimental procedures in order to make them more clear.
  • Figure 2 is missing some panels.
  • The authors' list is missing the surname of the second author.

Author Response

In this manuscript, Mitra and colleagues aimed at unveiling the effects of some molecules on the long-term survival/proliferation/NGF delivery capability of ECB devices composed of encapsulated NGC0211 cells. While I can found the aim of the manuscript interesting, I have to enlighten that most of the analyses here reported have been made on cells growing as monolayer in a condition that has limited relevance to the ECB device (3D growth). Notably, analyses performed on proper growth conditions (ECB, i.e. 3D conditions) have shown no major differences following the treatments considered, showing major difference with monolayer growth. 

Overall, this aspect limits the enthusiasm on this study. 

Response> We want to thank the reviewer for reading the manuscript thoroughly. In this study, we have compared the impact of various molecules on human mature hmNGF releasing cells (NGC0211) by growing the cells in different condition (2D and 3D culture) (hmNGF replaces NGF throughout the revised manuscript). Our main finding demonstrates that neither amyloid beta peptides not inflammatory molecules affect hmNGF release from the cells, or in other parameters including biochemical and cell death, when grown in either condition (2D and 3D culture). But these molecules affect the cell proliferation of NGC0211 cells in the short-term exposure set-up which we employed in our experiments. From these observations, we speculate that amyloid peptides and inflammatory molecules may similarly affect long-term efficacy of the NGC0211 cells. Our abstract clearly states the following ‘Aβ40/42, inflammatory molecules and activated astrocytes significantly reduced NGC0211 cell proliferation without affecting hmNGF release or other parameters. Long-term constant cell proliferation within ECB device is critically important to maintain a steady cell-population needed for stable hmNGF release, whereas hampered proliferation may lead to population decline thereby reducing hmNGF release.’ This portion of the abstract has been highlighted for your attention. Simultaneously, to exemplify the ‘prospective impact’ of our finding on long term ECB function, we slightly modified the title of our manuscript to clearly convey the manuscripts findings. In our future work, we will focus on finding the mechanistic basis on long-term impact on NGC0211 cells, as the reviewer suggested. 

Additional comments: 

  • Authors should revise experimental procedures in order to make them more clear. 

Response> Thank you for your comment. We have gone through the materials and methods section carefully and have incorporated changes wherever needed. 

  • Figure 2 is missing some panels. 

Response> We apologize for the mistake. All figures have been uploaded in the revised version. 

  • The authors' list is missing the surname of the second author. 

Response> We apologize for the mistake. We have now added the surname for second author. 

Reviewer 2 Report

In this study, the authors aimed to study the mechanisms regulating mNGF release from Encapsulated Cell Biodelivery (ECB) devices containing a specific engineered mNGF releasing cell line (NGC0211), in the context of Alzheimer’s disease (AD). Unfortunately, the study does not provide sufficient insights about the mechanisms affecting mNGF release by NGC0211 cells in vitro or within the ECBs. Indeed, the only clear effect mediated by direct or indirect (ACM) Aβ40/Aβ42 treatments is the increased reduction of Ki67 expression, which does not affect log-term NGF release from NGC0211 cells in the ECB devices, as claimed in the manuscript's title. Moreover, the overall statistical analyses are inadequate, and therefore the conclusions are not clear or properly supported.

Major comments:

Statistical analyses should be revised throughout the manuscript. In most cases, the applied statistical analyses based on one-way ANOVAs are not adequate for the intended comparisons and the underlying experimental design. For instance, in almost all experiments (except those included in Fig. 1), the authors draw conclusions by comparing the differential effects of Aβ40/Aβ42, treatment concentrations, different time points, etc., but these are not considered as variables in the analyses. In these cases, two-way or even three-way ANOVAs (Aβ species, concentration, and treatment duration) are required to stablish such comparisons.

Figure 2 A-D is missing.

NGF release from the ECB devices is not affected by direct or indirect exposure to Aβ40/Aβ42 (Fig. 6), because the observed reduction/increase occur also in the control conditions. Therefore, the results show no correlation between reduced Ki67 expression and long-term NGF release from the ECB devices, as claimed in the manuscript's title.

The effect of Aβ-induced astrocyte conditioned media on Ki67 expression (Fig. 5) is not convincing, since it may be mediated by Aβ oligomers and aggregated Aβ species remaining in the obtained conditioned media (after 24h of astrocyte conditioning with Aβ).

A control condition comprising treatment of NGC0211 cells with CSF from control/young subjects (non-cognitive impaired) is required in Fig.1B to assess the real effect of patients-derived CSF. Correlation of NGF release with t-tau, and if available phospho-tau levels, should be included.

Minor comments:

In Fig. 1D, independent regressions for each patient group should be indicated.

The authors must at least discuss the potential contribution of microglial cells and pathological tau species.

Legends of Supplementary Figure 5 (A) and (B) are switched.

Figure 6A is confusing: it seems that only two ECBs were analyzed in each condition, however the figure legend indicates “Data are represented as mean ± S.E (N = 3).”

Author Response

Reviewer 2

In this study, the authors aimed to study the mechanisms regulating mNGF release from Encapsulated Cell Biodelivery (ECB) devices containing a specific engineered mNGF releasing cell line (NGC0211), in the context of Alzheimer’s disease (AD). Unfortunately, the study does not provide sufficient insights about the mechanisms affecting mNGF release by NGC0211 cells in vitro or within the ECBs. Indeed, the only clear effect mediated by direct or indirect (ACM) Aβ40/Aβ42 treatments is the increased reduction of Ki67 expression, which does not affect log-term NGF release from NGC0211 cells in the ECB devices, as claimed in the manuscript's title. Moreover, the overall statistical analyses are inadequate, and therefore the conclusions are not clear or properly supported.

Response> We want to thank the reviewer for thorough reviewing of our manuscript. We want to specify that this study was conducted to evaluate the effect of factors (amyloid peptides, inflammatory molecules and activated astroglia) on NGC0211 cells, and we did not really intend to study the mechanisms behind the effect. This has been mentioned in introduction section 3rd paragraph last line (highlighted). This manuscript provides some of the preliminary data towards understanding the various effects of amyloid peptides and inflammatory molecules on NGC0211 cells. As mentioned in the introduction section, several pathways may be modulated in NGC0211 cells by various molecules present in the AD brain when the ECB devices are implanted. Thus, to understand the overall impact of several molecules, the present study was conducted. Due to astronomical number of molecules which can exert some effect on NGC0211 cells, we restricted our study only to amyloid beta peptides, inflammatory molecules and activated astrocytes, all of which are known to be quantitatively increased in AD. From our study, we could deduce that NGC0211 cells can resist the insult of all the molecules tested and ensure a steady NGF release for the limited duration tested in our study. But the significant impact on cellular proliferation by all the molecules tested signify that this is one of the target pathways which we should focus in our future studies. In accordance with this observation, we mentioned in our abstract, that the impact on cell proliferation ‘may lead to population decline thereby reducing hmNGF release’ (hmNGF replaces NGF throughout the revised manuscript). We have now made minor changes in the title of the manuscript to make this prospective impact of failure in cell proliferation more evident and clearer.

Major comments:

Statistical analyses should be revised throughout the manuscript. In most cases, the applied statistical analyses based on one-way ANOVAs are not adequate for the intended comparisons and the underlying experimental design. For instance, in almost all experiments (except those included in Fig. 1), the authors draw conclusions by comparing the differential effects of Aβ40/Aβ42, treatment concentrations, different time points, etc., but these are not considered as variables in the analyses. In these cases, two-way or even three-way ANOVAs (Aβ species, concentration, and treatment duration) are required to stablish such comparisons.

Response> We thank the reviewer for his/her observation on the statistical analysis. We went through the figures and statistical analysis and changed the way of comparing the included variances in each experiment. We reanalyzed the data and only compared them to the respective control subject in each experiment (or specific timepoint in case of kinetic studies). It has been mentioned throughout the manuscript ‘…….. as compared with their respective controls’. Thus, one-way ANOVA has been used in most of the figures.

Regarding the figure 4A-B, we apologize for missing to mention that two-way ANOVA analysis had been used in these experiments during the initial submission. Now we have included this statement both in the figure legend and in the section titles with statistical analysis. Additionally, in figure 6, since we compared between the pre-exposure and post-exposure levels of hmNGF release from the same ECB, the significance starts were taken off. We apologies for putting in the wrong analysis during the initial submission, but has been corrected now in the revised version.

Figure 2 A-D is missing.

Response> We apologies for the mistake. All figures have been uploaded in the revised version.

NGF release from the ECB devices is not affected by direct or indirect exposure to Aβ40/Aβ42 (Fig. 6), because the observed reduction/increase occur also in the control conditions. Therefore, the results show no correlation between reduced Ki67 expression and long-term NGF release from the ECB devices, as claimed in the manuscript's title.

Response> We thank the reviewer for pointing out this crucial observation. This is one of the most important observation in this manuscript that neither amyloid peptides nor ACM’s affect hmNGF release or cell viability of the NGC0211 cells when grown within the ECB, for the time frame utilized in this study. Instead, ACM’s increase hmNGF release and metabolic activity of the cells post-exposure. We have highlighted this observation in result section 3.6 last line, and in the discussion 5th paragraph (highlighted) and have mentioned in the abstract (highlighted). We go on to discuss possible reasons for this observation under the discussion section 5th paragraph. As discussed in the abstract (highlighted) and in the 4th paragraph of the discussion, we speculated that the severe anti-proliferative impact of amyloid peptides, inflammatory molecules and astroglial activation may eventually lead to population decline within the ECB device over time (possible long-term impact!). Similarly, when figure 2(E-H) and 3 is considered, we see hampered ki67 expression without alteration in hmNGF release as well, signifying that under the short time duration used in this study, hmNGF release is not significantly affected. We have made minor changes in the manuscript title to reflect this probable prospective impact of reduced cell proliferation on ECB long-term function more clearly.

The effect of Aβ-induced astrocyte conditioned media on Ki67 expression (Fig. 5) is not convincing, since it may be mediated by Aβ oligomers and aggregated Aβ species remaining in the obtained conditioned media (after 24h of astrocyte conditioning with Aβ).

Response> Thank you very much for this interesting comment. We are aware of the propensity of oligomeric amyloid peptides to form fibrils under in-vitro condition. Previous studies, performed by individuals related to our group, have shown that oligomeric amyloid peptides remain as oligomers when diluted in cell culture media for at least until 70hr (PMID: 21179413)(figure presented in the right). Since, our studies have used the same amyloid peptides, from the same supplier, followed the same protocol for suspension and are completed within 72hr timeline, we believe that the problem of fibrillar amyloid peptides is safely avoided. Furthermore, we used both direct exposure and indirect exposure (using ACM’s) to clearly show the difference between them and to highlight the specific effects of activated astrocytes on the NGC0211 cells. Thus, we believe that our observations on ki67 expression is purely from the effect of oligomeric amyloid peptides, either directly or through the activation of astrocytes (ACM’s). To avoid confusion, we have deleted the probability of oligomeric amyloid beta to form fibrillar amyloid beta from the discussion section (2nd paragraph, first sentence). We have also added appropriate text and reference under method section 2.3 (highlighted).

A control condition comprising treatment of NGC0211 cells with CSF from control/young subjects (non-cognitive impaired) is required in Fig.1B to assess the real effect of patients-derived CSF. Correlation of NGF release with t-tau, and if available phospho-tau levels, should be included.

Response> We thank the reviewer for this comment. Unfortunately, we presently do not have control CSF from other condition. It would take few months to acquire such samples, and as the reviewer may generously acknowledge, it is very difficult and ethically challenging to acquire such samples. Unfortunately, we also want to state that the correlation data with t-tau and phospho-tau are presently communicated as part of another publication (different manuscript), and thus we cannot use those data in the present manuscript.

For reviewer’s eye only, we are providing the p-tau levels here (figures on the right). The data shows the presence of p-Tau in the human CSF sample used in this study along with the ratio between p-Tau/Aβ42.

Minor comments:

In Fig. 1D, independent regressions for each patient group should be indicated.

Response> Thank you for the comment. Fig 1D has now been modified to show independent regression for each patient group.

The authors must at least discuss the potential contribution of microglial cells and pathological tau species.

Response> Thank you for the comment. We have now added a section in the 3rd paragraph under the discussion section.

Legends of Supplementary Figure 5 (A) and (B) are switched.

Response> We apologize for the mistake. We have now switched the figure and made appropriate changes in the manuscript.

Figure 6A is confusing: it seems that only two ECBs were analyzed in each condition, however the figure legend indicates “Data are represented as mean ± S.E (N = 3).”

Response> We apologize for the confusing presentation of data. We have compared the same ECB – pre and post exposure and only two ECB devices was included for this experiment which are shown in the figure. Now we have rectified this mistake in the figures and figure legend. We also have taken away the statistical analysis.

Reviewer 3 Report

The paper by Mitra et al. submitted to "Cells" reports on the encapsulation of nerve growth factor (NGF), the cellular and release responses in long-term and the putative use for therapy in Alzheimers disease (AD). They show that NGC0211 are resistent to Ab40/42 peptides and astroglia-induced stress but show significant anti-proliferative activity possibly affecting the maintenance of NGC0211 cells. The paper comes from a well known group (Maria Eriksdotter) working extensively on this topic, which even came into a clinical trial. The authors go on and aim to optimize their system. All methods are well done and established. There are a few minor and major critics which should be considered.

  1. I am not really happy with the abbreviaton mature "mNGF". First it is not fully consistent throughout the whole MS. Second, sometimes it is also used for mouse/murine NGF. Also it should read human mature (hm) NGF, although there is sequence homology. In line 245 it says NGF or beta-NGF.

  1. In Introduction the p75NTR receptor should at least be mentioned.

  1. The authors use a 280 kDa cut-off ECB? Why as NGF is rather a small molecule of 14 kDa? Comment please. In this regard the statement in line 503, that Ab peptides (4kDa) can potentially diffuse inside the ECB is irrelevant as everything <280 kDa can diffuse inside.

  1. On line 89, the authors mention that the new system allows 10x more NGF production (give a value here ng/ml per 24hr per 1 million cells).

  1. In general I am not happy about the raw values of NGF release, it is only partly correct in Figure 6, where it reads ng/mL per 24hr (possibly also better per 1 million cells). In Fig.1B it says only ng NGF, which is wrong (or at least ng/mg total protein if related to tissue), and in Fig.2 E only ng/mL. This is not consistent and unclear.

  1. A major issue on this story is the application of beta-amyloid; here I am missing that it is human species. I also would like to see that it is really a 4kDa monomer in your hands and not an aggregate. Show it in a Western Blot. In this respect it will be highly interesting to see also effects of aggregated Ab42 (which is easy to do and even more potent). This is also important to show as the Ab peptides can spontanously aggregate during the culturing with NGC cells; this should be tested and discussed. I do not think that you will have a 4kDa peptide throughout the whole experiments, at least small aggregates may occur.

  1. What about the ELISA, give the detection limit and mention that it does not bind pro-NGF.

  1. You mention the analysis of CSF tau and bA; in fact also pTau181 is state-of-the-art. Comment on that please.

  1. Partly the significance stars (*) are not easy to read; a typical example is Figure2F&G (very good to read) and Figure 2E&K (hard to see) and very bad in Figure 4A.

  1. I cannot find the antibodies S100 and GFAP in methods (even if supplementary)

11.One serious concern, is that you have not included CSF of "healthy controls" or "cognitively not impaired controls" or another "alternative normal control". I know this is difficult but the only way to compare correctly.

  1. The authors should discuss the potential use of CSF in their experiments. Give CSF levels of Ab42 (approx. 800 pg/ml) and Ab40 (approx. 10,000 pg/ml) in CSF and discuss if such low concentration can really have a potential effect on your NGC cells. I guess the concentrations are too low to affect anything. Comment please.

Author Response

Reviewer 3

The paper by Mitra et al. submitted to "Cells" reports on the encapsulation of nerve growth factor (NGF), the cellular and release responses in long-term and the putative use for therapy in Alzheimer’s disease (AD). They show that NGC0211 are resistant to Ab40/42 peptides and astroglia-induced stress but show significant anti-proliferative activity possibly affecting the maintenance of NGC0211 cells. The paper comes from a well-known group (Maria Eriksdotter) working extensively on this topic, which even came into a clinical trial. The authors go on and aim to optimize their system. All methods are well done and established. There are a few minor and major critics which should be considered.

Response> We want to thank the reviewer for critical understanding of the present work. Indeed, the NGC0211 cells showed resistance to amyloid peptides and astroglia induced stress, except showing its (NGC0211 cells) vulnerability to the anti-proliferative impact from amyloid peptides and astroglia. We believe the anti-proliferative impact on NGC0211 cells may lead to altered ECB function eventually. 

  1. I am not really happy with the abbreviation mature "mNGF". First it is not fully consistent throughout the whole MS. Second, sometimes it is also used for mouse/murine NGF. Also it should read human mature (hm) NGF, although there is sequence homology. In line 245 it says NGF or beta-NGF.

Response> Thank you for the comment and suggestion. We have now changed all reference of human mature NGF to hmNGF, specifically those which are related to the ECB mediated delivery and NGC0211 cells. We kept all mention of mature NGF as mNGF, only when mature NGF is referred in a neutral context (irrespective of species, basic biology context), for example – explaining the background information about mature NGF’s biological activity and effect on BFCN’s. To reduce the confusion, we deleted the name of the kit, used to measure hmNGF from the materials section, kept only the catalogue number and company information, and have made appropriate changes in the methods section.

  1. In Introduction the p75NTR receptor should at least be mentioned.

Response> We thank the reviewer for this comment. p75NTR receptor has been mentioned in the introduction section appropriately.

  1. The authors use a 280 kDa cut-off ECB? Why as NGF is rather a small molecule of 14 kDa? Comment please. In this regard the statement in line 503, that Ab peptides (4kDa) can potentially diffuse inside the ECB is irrelevant as everything <280 kDa can diffuse inside.

Response> We thank the reviewer for this comment. It is true that NGF is a small molecule. The ECB-platform has been optimized using 280kDa cut-off membrane and had been used to deliver several different neurotrophic molecules in the past, including the previous clinical trial for ECB-NGF in AD patients. Thus, we used the optimized device configuration for this study as well, to correlate our finding with previous and other ongoing studies. It is true that any molecule <280kDa can diffuse inside the device, but since this study is focused on amyloid peptides, inflammatory molecules and activated astrocytes, we mentioned this explicitly in introduction and discussion sections.

  1. On line 89, the authors mention that the new system allows 10x more NGF production (give a value here ng/ml per 24hr per 1 million cells).

Response> Thank you for the comment. We have added the values in the introduction section.

  1. In general I am not happy about the raw values of NGF release, it is only partly correct in Figure 6, where it reads ng/mL per 24hr (possibly also better per 1 million cells). In Fig.1B it says only ng NGF, which is wrong (or at least ng/mg total protein if related to tissue), and in Fig.2 E only ng/mL. This is not consistent and unclear.

Response> Thank you for the observation. We have made appropriate changes in calculations and corrected the labeling to make it consistent as : hmNGF (ng/ml/24hr) in all graphs. For figure 6, we cannot really state the value of hmNGF released upon the number of cells, and thus we denote the value released from individual devices instead.

  1. A major issue on this story is the application of beta-amyloid; here I am missing that it is human species. I also would like to see that it is really a 4kDa monomer in your hands and not an aggregate. Show it in a Western Blot. In this respect it will be highly interesting to see also effects of aggregated Ab42 (which is easy to do and even more potent). This is also important to show as the Ab peptides can spontaneously aggregate during the culturing with NGC cells; this should be tested and discussed. I do not think that you will have a 4kDa peptide throughout the whole experiments, at least small aggregates may occur.

Response> Thank you very much for this comment. We have now mentioned in the manuscript about the details of the amyloid peptides under methods section 2.3. In the same method section, we mentioned that the amyloid species we used in this manuscript are oligomers and not specifically monomers.

We and other collaborators in the department had previously published about the oligomeric form of the amyloid peptides we purchase from rPeptides. Specifically, in a previous publication, Hjorth et al., (PMID: 23481688) showed that the oligomeric amyloid peptides are primarily monomer and dimers (shown in figure in the right, panel A).

Simultaneously, it has been shown previously (PMID: 23481688), that when dissolved in culture medium, the oligomeric properties do not change significantly as shown below (panel B)

Along with maintaining the oligomeric properties under culture conditions, our collaborators using thioflavin T assay have shown that fibrillation of oligomeric amyloid peptides does not occur under culture condition at least until 70hr (PMID: 21179413) (please refer figure on the right).

Since our studies have used the same amyloid peptides, from the same supplier, followed the same protocol for suspension and are completed within 72hr timeline, we believe that the problem of fibrillar amyloid peptides is safely avoided.

We agree with the reviewer that it would be interesting to check the effect of fibrillar or even aggregated amyloid peptides on NGC0211 cells, which we will address in our future experimental studies.

  1. What about the ELISA, give the detection limit and mention that it does not bind pro-NGF.

Response> We thank the reviewer for this comment. We have now added the assay range of the ELISA in materials and methods section 2.9.

As stated by the manufacturer, this assay detects an immunogen for mature NGF, which means that this assay would also detect proNGF as well, since it contains the same sequence. But to avoid misinterpretation of our data, and to avoid the interference from proNGF especially in ACM’s we also showed the amount of basal NGF production by astrocytes in supplementary figure 3B. This observation has been already discussed in 5th paragraph of the discussion section (highlighted). Since the amount of NGF (pro or mature) produced by astrocytes is negligible compared to the amount of hmNGF produced by the NGC0211 cells, we believe it would not alter our data’s interpretation. In case of NGF estimation released by NGC0211 cells, they release only hmNGF and thus there should be no involvement of pro-NGF in the ELISA assay.

  1. You mention the analysis of CSF tau and bA; in fact also pTau181 is state-of-the-art. Comment on that please.

Response> Thank you for the comment. We agree with the reviewer that p-Tau content is state-of-the-art in classifying AD cases from other types of dementia. Unfortunately, the correlation data with t-tau and phospho-tau are presently communicated as part of another publication (different manuscript), and thus we cannot use those data in the present manuscript.

For reviewer’s eye only, we are providing the data here (figures on the right). The data shows the presence of p-Tau in the human CSF sample used in this study along with the ratio between p-Tau/Aβ42.

  1. Partly the significance stars (*) are not easy to read; a typical example is Figure2F&G (very good to read) and Figure 2E&K (hard to see) and very bad in Figure 4A.

Response> We thank the reviewer for pointing this out. We have now corrected this issue.

  1. I cannot find the antibodies S100 and GFAP in methods (even if supplementary)

Response> Now this is included under materials and methods section in supplementary and also in figure legends of the supplementary figure 2.

  1. One serious concern, is that you have not included CSF of "healthy controls" or "cognitively not impaired controls" or another "alternative normal control". I know this is difficult but the only way to compare correctly.

Response> We thank the reviewer for this comment. We agree that it would have been ideal to use a control CSF in our study. Unfortunately, we presently do not have control CSF from other condition. This is also the reason why we used AD CSF along with other condition (LBD and SCI) in our manuscript, to see whether disease severity or other dementia condition have any differential effect on NGC0211 cells. We hope the reviewer may generously acknowledge that it would take few months to acquire such samples, which are otherwise difficult to acquire and sometimes ethically challenging.

  1. The authors should discuss the potential use of CSF in their experiments. Give CSF levels of Ab42 (approx. 800 pg/ml) and Ab40 (approx. 10,000 pg/ml) in CSF and discuss if such low concentration can really have a potential effect on your NGC cells. I guess the concentrations are too low to affect anything. Comment please.

Response> We thank the reviewer for this comment. We agree with the reviewer’s comment that the levels of amyloid peptides in CSF were in the low nanogram range, and according to our in-vitro experiments (fig 2)(range used 1 - 0.05 µg) , the amyloid peptides may not really be the reason behind reduced hmNGF release shown in figure 1. We had already explicitly mentioned this aspect in the 1st paragraph of discussion (highlighted). Moreover, we have now added another sentence in results section 3.1 (highlighted) to make this point clear. We carry on our work to identify the potential factor present in the diseased CSF which is found to repress hmNGF release from NGC0211 cells in our future studies.

Round 2

Reviewer 1 Report

The revised version of the manuscript has been improved with respect to the original submission. The authors have worked on the text in order to fix some overextimations of their findings. The text contains some typos and I suggest to make work on the figures in order to make them more clear and immediate.

Author Response

Revision 2

The revised version of the manuscript has been improved with respect to the original submission. The authors have worked on the text in order to fix some overestimation of their findings. The text contains some typos and I suggest to make work on the figures in order to make them more clear and immediate.

> We earnestly thank the reviewer for your time and effort which helped us to improve our manuscript considerably. We have now checked through the manuscript carefully. We changed text, figures, and figure legends to make the manuscript easy to read and comprehend. Based on the 2nd reviewers’ comments, we have now changed the title of the manuscript to explicitly convey the work performed in ‘this manuscript’ exclusively. We sincerely hope that the changes made would complete the manuscript and make it ready for acceptance.

Reviewer 2 Report

The manuscript has not been sufficiently improved and most of my major concerns remain. A manuscript's conclusions and title cannot be speculative, and must limit to the obtained results; therefore, using “this observation may affect something that was not assessed” in a title is not appropriate.

Author Response

Revision 2

The manuscript has not been sufficiently improved and most of my major concerns remain. A manuscript's conclusions and title cannot be speculative, and must limit to the obtained results; therefore, using “this observation may affect something that was not assessed” in a title is not appropriate.

> We are very sorry to know that all the changes we made during the first revision did not fully address the reviewer’s queries. We have made considerable effort to address all the queries again. We have now consulted with experts in statistical analysis, and they recommend that the one-way ANOVA used in our experiments are of publishable quality, since each experiment contains its individual control at all given time-point, provided we do not compare between different treatment groups. Although much more statistical analysis can be made, but to observe changes with respect to treatments, we strictly showed changes in comparison to control samples at each time point. Thus, we reframed to compare between the treatment groups themselves. Moreover, treatments do not always lead to linear outcomes, which further complicates the issue (with so many time points and concentrations), which we have tried to avoid otherwise it would severely complicate the write-up in the results section.

Regarding the speciation of the amyloid beta peptides, we already provided considerable data and proof during the first revision. Moreover, we have now changed the title of the manuscript, to specifically reflect the content of this manuscript.

We earnestly believe that the changes made are of highest publishable quality, which had now been consulted with the experts in the field. We would be happy to address any further queries from the reviewer which he/she may specify during further correspondence.

Reviewer 3 Report

The revision responds to all my concerns. I suggest to accept the MS.

Author Response

Revision 3

The revision responds to all my concerns. I suggest to accept the MS.

> We want to thank the reviewer for your kind consideration for acceptance of our manuscript. We really appreciate your time and effort for enhancing the quality of our manuscript.